# Studies on Glass Fiber-Reinforced Poly(Ethylene-Grafted-Styrene)-Based Cation Exchange Membrane Composite

**DOI:** 10.3390/ma13245597

**Published:** 2020-12-08

**Authors:** Di Huang, Zhichao Chen, Jiann-Yang Hwang

**Affiliations:** 1Department of Materials Science and Engineering, Michigan Technological University, Houghton, MI 49931, USA; 2Evergreen Technology Inc., Hancock, MI 49930, USA; zhichaoc@mtu.edu

**Keywords:** cation exchange membrane, glass fiber, coupling agent, tensile strength, tensile modulus

## Abstract

To improve interfacial adhesion between glass fiber (GF) and poly(ethylene-grafted-styrene)-based cation exchange membranes (CEM), GF was modified by four coupling agents: [3-(Methacryloxy)propyl] trimethoxy silane (3-MPS), 1,6-bis (trimethoxysilyl) hexane (1,6 bis), Poly(propylene-graft-maleic anhydride) (PP-g-MA) and Triethoxyvinylsilane (TES). The results indicated the addition of modified GF increased tensile strength, tensile modulus, storage modulus and interfacial adhesion of GF/CEM composite but degraded the strains. The composite with 3-MPS modified GF obtained superior mechanical properties and interfacial adhesion, whereas the modified effect of TES was inconspicuous. The addition of unmodified GF even had negative effects on GF/CEM mechanical properties. The field emission scanning electron microscopes (FE-SEM) showed that the GF treated by 3-MPS and PP-g-MA have better compatibility with the CEM matrix than 1,6 bis and TES-treated GF. The Fourier-transform infrared spectroscopy (FT-IR) verified that the strengthening effects from modified GF were attributed to the formation of Si-O-Si and Si-O-C bonds. The additions of modified GF in CEM positively influence water uptake ability but negatively influence ion exchange capacity (IEC). This research provided a way of strengthening GF/CEM composite and pointed out which functional groups included in coupling agents could be useful to GF-reinforced composite.

## 1. Introduction

Over recent decades, there has been renewed interest in developing high-performance ion exchange membranes, which play important roles in various industrial applications, such as fuel cell, desalination and wastewater treatment [1,2,3,4,5]. Typically, ion exchange membranes are composed of inert polymers such as polyethylene, polyvinylidene fluoride and polyvinylchloride [6,7,8,9], and reactive polymers such as polystyrene and polysulfone that can be functionalized by ionic groups [8,10]. The desired properties of ion exchange membranes are determined by the inert backbone polymers, reactive polymers and methods to combine them.

According to the connecting way of charged groups on the matrix, ion exchange membranes can be classified into homogeneous and heterogeneous membranes [11]. Homogeneous ion exchange membranes have wide applications due to the excellent electrochemical performance [11]. However, during the long service life, homogenous ion exchange membranes cannot maintain the desirable mechanical properties and structural stability in the harsh environment, such as plating wastewater, which requires the high structural and chemical stability for the membranes [12]. The studies of improving mechanical properties and microstructure stability of ion exchange membranes have attracted much attention in recent years. Mechanical property is a key factor for ion exchange membrane in electrodialysis (ED) stack application. Although the ED is not a pressure driven process, the membrane needs to resist some small overpressure during operation.

Glass fiber (GF)-reinforced polymer composite gradually became a competitive structural material due to the excellent mechanical properties. Three factors affect the fiber-reinforced composite’s mechanical properties: intrinsic properties of matrix materials, the strength of interfacial chemical bonding between the fillers and matrix and the load transfer efficiency of interphase [13]. A large amount of research indicated if GF was not treated by modifier, the mechanical improvement effects of GF on composite is confined [14,15]. This is due to the insertion of GF, which destroys the phase morphology of the matrix. GF was also reported to have lower impact resistance and load transferability of the fiber-reinforced composite due to hydrophilic nature of hydroxyl groups. The reason behind that is that no chemical bonds existed between the GF and matrix.

Recently, many methods have been studied to improve the interfacial adhesion and construct chemical bonds between the fibers and matrix. The styrene/2-ethylhexylacrylate/divinylbenzene solid foams was reported to be reinforced by the sonicated silica particle up to 3%. The sonication-treated silica particle considerably enhanced the composites’ Young’s modulus and crush strength [16]. Zhang et al. reported the mechanical properties of long glass fiber-reinforced polypropylene (PP) improved using dicumyl peroxide and maleic anhydride as adhesion promoters to increase interfacial interaction between PP and GF. This research obtained the result that the content of dual compatibilizer could influence the composite’s strengthening extent [17]. Nano-silica treated silane coupling agents, including γ-aminopropyl-triethoxysilane (KH550), Vinyl-triethoxysilane (A-151) and γ-chloro-propyl trimethoxy silane (A-143), were reported to modify GF and could effectively improve the composite’s mechanical properties [13]. This study also concluded that the coupling agent’s content affected the GF-reinforced composite’s mechanical property enhancement. The γ-(methacryloyloxyethyl) trimethoxy silane has been reported to treat the GF and lead to a vinyl functionalized GF surface, which served to covalently bond between styrene/divinylbenzene copolymer and GF [18]. Other interfacial adhesion strengthening methods were studied including introducing adhesion prompters to increase the compatibility between the fiber and matrix [19,20]. Compatibilizers such as admicellar, maleated ethylene and a few acrylic acid copolymers were also reported to enhance the adhesion between the fiber and matrix [19,21]. In above studies, treating GF with the adhesion promoter and compatibilizer are a promising way to improve interfacial adhesion between the GF and matrix.

The idea of improving interfacial adhesion between the fiber and polymeric matrix by modifying the fiber surface can also be introduced to GF and ion exchange membrane composite. In Křivčík’s work, the polypropylene and GF were used as a low-cost alternative to common woven fabric in ion exchange membrane [22]. He confirmed that short polypropylene fibers rapidly increased the mechanical strength in machine direction. Annala et al. reported that the storage modulus and loss modulus of sulfonated ethylene/styrene copolymer were improved moderately after adding GF [23]. The strengthening effect of GF on composite was confined without surface modification. However, studies of improving the mechanical property of GF/ion exchange membrane composite through modifying the GF’s surface have not been widely concerned. Most studies focused on even dispersing GF on the matrix, rather than improving interfacial adhesion, such as the above two examples. The issue is that the introduction of the higher modulus GF leads to the increase in tensile strength but destroys the phase morphology of the matrix [14]. The damage of the matrix’s consistency would result in the resistance to the composite strengthening. In our study, four coupling agents were chosen as surface modifiers to improve the interfacial adhesion. In addition to the silane surface modifiers (3-(Methacryloxy)propyl] trimethoxy silane (3-MPS), 1,6-bis (trimethoxysilyl) hexane (1,6-bis) and Triethoxyvinylsilane (TES)), the compatibilizer (Poly(propylene-graft-maleic anhydride) (PP-g-MA)) was also chosen to study the treatment effect on the GF surface. TES-treated GF was reported to improve the composite’s mechanical properties effectively [24]. However, the GF was not directly treated by TES. It was combined with nano-silica to modify GF. PP-g-MA was reported to be combined with two organofunctional silanes to increase the interfacial adhesion in glass fiber-polypropylene (PP)-reinforced composites [25]. The 3-MPS and 1,6 bis contained one and two trimethoxy, respectively; the characteristic functional group in many coupling agents for GF modification was chosen to study the treatment effects due to analogous organofunctional structure.

The objective of this article is to investigate the modifying effects of different coupling agents on GF, which were used to strengthen poly(ethylene-co-styrene)-based cation exchange membrane. The mechanical properties, morphology characterization and FT-IR of GF-reinforced poly(ethylene-grafted-styrene)-based cation exchange membrane (CEM) composites were studied in this paper. To assure the mechanical property enhancement is not at the cost of other membrane properties, the ion exchange capacity and water uptake ability were also investigated in this paper.

## 2. Experimental

### 2.1. Materials

1,6 bis was supplied by Gelest Inc; 3-MPS and PP-g-MA (maleic anhydride 8–10%) were supplied by Sigma Aldrich Co, Ltd. St. Louis, MO, USA); TES was purchased from TCI Co, Ltd. (Portland, OR, USA); low density polyethylene films (LDPE, 102 μm) were supplied by Uline Co. Ltd. (Pleasant Prairie, WI, USA). GF was supplied by Fibre Glast Developments Corporation (Brookville, OH, USA), with a diameter of sixteen microns and average length of two hundred and thirty microns. Styrene (99.9% purity), divinylbenzene (DVB, 80% purity), Luperox A98 (benzoyl peroxide, 98%), 1,2-dichloroethane, chlorosulfonic acid and polystyrene-block-poly-(ethylene-ran-butylene)-block-polystyrene were purchased from Sigma Aldrich Co, Ltd. (St. Louis, MO, USA). Toluene, xylene and hydrochloric acid (HCl, 37% purity) were supplied by Carolina Chemical Co, Ltd. (Charlotte, NC, USA). All solvents were used as received.

### 2.2. Surface Treatment of GF

The coupling agents were hydrolyzed in ethanol alcohol solution, adjusting pH to 3.5 using diluted hydrochloric acid. The concentration of coupling agents in ethanol solution was 2%. After 1 h hydrolyzation, GF was immersed in the different coupling agent solutions for 1 h at 100 °C. Then, the glass fibers were dried at room temperature for 24 h. The chemical structure and hydrolyzation mechanism of these four coupling agents are illustrated in Figure 1.

When the effect of the coupling agent’s concentration on the mechanical properties of composite was investigated, the concentration in ethanol solution varied to 1%, 3%, 4% and 5%, respectively.

### 2.3. Synthesis of Poly(Ethylene-Grafted-Styrene) Copolymer(PE-g-PS) and GF/PE-g-PS Membrane Composite

LDPE films were rinsed with acetone and dried at 50 °C for 6 h to remove moisture. Then, 2 g LDPE was dissolved in the mixture composed of 20 mL toluene and 2 mL xylene at 90 °C. After LDPE was completely dissolved, the styrene, DVB and benzoyl peroxide were added to the above mixture in the nitrogen atmosphere. The reaction was kept at 90 °C for 6 h. The mass ratio between PE and styrene was 1:1.5. The DVB and benzoyl peroxide were added at a weight ratio of 3% and 0.5% to styrene. When the reaction was completed, the copolymer was stirred with the GF at 200 °C for 20 min. The formulation of composite with untreated GF, treated GF and GF treated by different coupling agent concentrations is summarized in Table 1. The thermal plastic elastomer, polystyrene-block-poly-(ethylene-ran-butylene)-block-polystyrene, was blended with the GF/PE-g-PS composite to increase the ductility. The viscous composite was then cast onto a piece of glass and slicked by a hot press machine to make the GF evenly distributed on the membrane. All the composites were carefully prepared under the same processing condition as above. The whole procedure is shown as Scheme 1.

### 2.4. Sulfonation of GF/PE-g-PS Composite for Cation Exchange Membrane Preparation

Sulfonation of GF/PE-g-PS membrane composite was carried out by immersing it in the chlorosulfonic acid solution (5% chlorosulfonic acid mixed with 95% 1,2-dichloroethane) at 0 °C for 2 h. After sulfonation, the resulting membrane composite was repeatedly rinsed with distilled water to remove residual chlorosulfonic acid, and the cation exchange membrane composite was obtained after drying in an oven at 50 °C overnight.

### 2.5. Sample Characterization

Mechanical properties. The evaluation of mechanical properties to GF/CEM composites includes dynamic mechanical analysis (DMA) and tensile strength tests. These tests can be obtained from the DMA Q 800 machine (TA Instruments, New Castle, DE, USA). The DMA provides storage modulus and loss factor, being monitored from 25 to 150 °C, at a heating rate of 3.00 °C·min^−1^. Tensile strength was tested at the loading rate of 0.5000 N·min^−1^ until 5.0000 N.Morphology characterization. The microstructure, compatibility of GF and the matrix membrane and phase separation of composites were characterized by FE-SEM analysis. The Hitachi S-4700 FE-SEM (Schaumburg, IL, USA) is equipped with an Oxford energy dispersive X-ray spectroscopy (EDS) microprobe (Abingdon, UK).Fourier-transform infrared spectroscopy (FT-IR) analysis. The functional groups of the matrix membranes and final composite were investigated by FT-IR. A PerkinElmer Spectrum One FT-IR spectrometer (Chicago, MI, USA) with a universal attenuated total reflection accessory was used to record the infrared spectrum.Ion exchange capacity (IEC). The IEC of CEM was determined by the titration method. The dried membranes were cut into small pieces and stirred with 1.0 M sulfuric acid solution overnight to make sure sulfonate groups are in H^+^ form. Then, the membranes were washed with distilled water to remove the excess sulfuric acid. The resulting membranes were dried in the oven at 60 °C and then stirred with 50 mL 0.5 M NaCl solution overnight. Due to the replacement of H^+^ on the membrane by sodium cation from the solution, the solution becomes acidic. The concentration of H^+^ was determined by pH measurement and titration using a diluted solution (e.g., 0.01 M KOH). The IEC values were calculated using the Equation (1):
IEC = (V_KOH_ × C_KOH_)/W_dry_(1)
where, V_KOH_ is the volume of KOH used in the titration, and W_dry_ is the dry weight of the membrane in g. C_KOH_ is the molarity(mol/L) of KOH used in the experiment for the titration.Water uptake. The water uptake experiments were performed by measuring the weight differences between dried membranes and fully hydrated membranes. The membranes were cut into small pieces and immersed in deionized water at 25 °C with a designed duration. After that, the surface moisture was removed, and mass was weighed (W_wet_). The wet membrane was dried until the water evaporated completely and weighed again (W_dry_). The water uptake can be determined by Equation (2).
(2)$$\mathrm{Water}\;\mathrm{uptake}=\frac{W_{\mathrm{wet}}-W_{\mathrm{dry}}}{W_{\mathrm{dry}}}\times 100\%$$Swelling rate. The extent of swelling in membrane composite can be determined via the changes in linear dimensions of edge length. It can be calculated using the Equation (3):
(3)$$\mathrm{Swelling}\;\mathrm{rate}\;=\frac{L_{s}-L_{o}}{L_{o}}\times 100\%$$
where, L_s_ is the mean value of swollen membrane edge length, the L_o_ mean value of original edge length.

## 3. Results and Discussions

### 3.1. The Influence of Various Coupling Agents on Mechanical Properties of CEM Composite

The effects of various coupling agents on mechanical properties of GF/CEM composite are illustrated in Figure 2 and Figure 3. The pure membrane specimen is the membrane without GF. The blank sample is the ion exchange membrane with untreated GF. Compared to the virgin membrane and blank sample, the tensile strength of composites with GF treated by 1,6 bis, 3-MPS and PP-g-MA all increased. The sample treated by 3-MPS obtained significant stress enhancement. All the samples with treated GF have higher tensile modulus. This is also reflected by the slopes of stress–strain curves in Figure 2. The steeper the slope, the higher the tensile modulus. The improved tensile strength and tensile modulus results indicate that with the addition of GF, the CEM composites would not stretch easily and have better deformation resistance against water pressure. However, the strain value decreased when adding GF fillers in CEM, especially when adding the GF treated by coupling agents. Those results indicate that the GF/CEM composites become strong but not tough with modified GF. The composite with modified GF needs more force to be broken but cannot suffer higher elongation.

Not only the types of coupling agent affect composites’ mechanical strength but also the coupling agent’s concentrations. Since the 3-MPS showed a superior modification effect on GF, the 3-MPS modified GF was chosen to investigate the effect of coupling agent concentration on mechanical properties. Figure 4 exhibits the stress–strain relationship of CEM composites with the GF treated by increasing 3-MPS concentration. The results show that the 2% 3-MPS concentration had a superior treating effect on CEM composite’s stress enhancement. However, the strains weakened gradually by increasing 3-MPS concentration. Figure 5 shows that the tensile strength and tensile modulus of GF/CEM composite increased until the 3-MPS’ concentration reached 2%. After that, the tensile strength and tensile modulus both decreased with the increasing 3-MPS concentration treated on GF.

### 3.2. The Effect of Various Coupling Agents on Dynamic Mechanical Property of CEM Composites

The dynamic mechanical property can be evaluated by storage modulus and loss modulus. The storage modulus reflects the ability of materials storing energy elastically, representing the elastic portion. The loss modulus reflects the ability of materials dissipating stress through heat, representing the viscous portion. The loss modulus to storage modulus ratio in a viscoelastic material is defined as the tan delta, which provides a measure of damping in the material.

The influences of various coupling agents on storage modulus as a function of temperature are presented in Figure 6. It was found that the storage modulus of GF/CEM composites with 1,6 bis, 3-MPS, PP-g-MA modified GF were all improved compared to the blank sample and pure membrane, which explains why the tensile strength and tensile modulus enhanced. This effect was more noticeable when the temperature was below 120 °C, which can be attributed to interfacial chains between the GF and matrix that become soft when the temperature is high [26]. The unmodified GF, which means no adhesion between the GF and matrix, resulted in the negative influence on storage modulus at the higher temperature. This negative influence is because the membrane’s integrity was destroyed by GF insertion. The GF treated by TES did not show a positive effect on composite, resulting in lower storage modulus when the temperature was below 70 °C.

The loss factor (tan δ)–temperature curves in Figure 7 have two bumps corresponding to two transitions. The first transition happened around 65–70 °C, corresponding to the glassy-rubbery transition. The second transition, which occurred about 120–135 °C, is associated with crystallite transformation. The crystallites transformation represents the interface adhesion status of composite material. If briefly compared with the blank sample, the crystallite transformation bumps of CEM composites with modified GF shifted to a higher temperature range. This shift revealed the interfacial adhesion between the matrix and GF were improved after being treated by coupling agents. The tan delta also enhanced after GF modification, indicating that the ability of CEM composite to dissipate stress was improved. The fillers’ interfacial adhesive effects on dynamic mechanical properties can be accurately expressed by Luis Ibrarra’s formula [27]. This Equation (4) assumes that the composite loss factor is the sum of component loss factor and volume fraction products.
(4)$$\;\tan\delta c=V_{f}{\tan\delta }_{f}+V_{i}{\tan\delta }_{i}+V_{m}{\tan\delta }_{m}$$
tan δ_c_ and tan δ_m_ represent the loss factors for the composite and matrix, respectively. V_f_ represents the volume fraction of glass fibers. The parameter A is introduced to measure the adhesive effects between the fillers and matrix; A is defined as Equation (5):(5)$$A=\frac{V_{i}}{1-V_{f}}\times \frac{{\tan\delta }_{i}}{{\tan\delta }_{m}}$$

Assuming ${\tan\delta }_{f}=0$, the interfacial volume fraction V_i_ is ignored, then Equation (4) can be rewritten as Equation (6):(6)$$\frac{{\tan\delta }_{c}}{{\tan\delta }_{m}}=\left(1-V_{f}\right)\left(1+A\right)$$

Hence, the parameter A can be rewritten as Luis Ibrarra’s formula (Equation (7)):(7)$$A=\frac{1}{1-V_{f}}\times \frac{{\tan\delta }_{c}}{{\tan\delta }_{m}}-1$$

The A is an inverse proportion to the interfacial adhesion. Table 2 shows the A values and the corresponding tanδ_c_ and tanδ_m_. The A values of GF/CEM composites with GF treated by 3-MPS and PP-g-MA were relatively lower than others, indicating higher glass fiber–matrix adhesive effects obtained at their interface. 3-MPS and PP-g-MA were also demonstrated have better modification effects on improving the adhesive capacity of the interface. The negative value of A in the blank sample represents the addition of raw GF impaired interface adhesion. The applied force could not dissipate from the matrix to GF.

The effects of 3-MPS’s concentration on DMA as a function of temperature are shown in Figure 8. The composite with GF treated by 1% 3-MPS had the lowest storage modulus value in the temperature range 25–150 °C due to the insufficient modification effect. The 2% 3-MPS had the best modification effect on GF, making the GF/CEM composite obtain the largest storage modules. However, continually increasing the 3-MPS weakened this enhancement effect. The tensile stress and tensile modulus of composite with 3-MPS-treated GF had the same trend. A possible reason would be that the 2% 3-MPS concentration was enough to react with Si-OH on GF. The higher 3-MPS concentration would lead to the GF’s surface being covered by the redundant coupling agents, decreasing the GF and membrane matrix’s interfacial adhesion.

### 3.3. CEM Composites Morphology Analysis

The surface morphologies of CEM composites with untreated GF and GF treated by four coupling agents were investigated by FE-SEM (as shown in Figure 9 and Figure 10). Figure 9a shows that most GFs were covered under the membrane matrix, but some single glass fibers were dispersed out of the matrix. In addition, many holes appeared in the matrix due to the GF insertion. Figure 9b shows that the GF’s surface was smooth, and the cavity around GF was loose. This revealed that the interfacial adhesion was inferior due to the lack of chemical bonds between the GF and matrix. This weak adhesion resulted in low mechanical properties of the composites, and the holes caused by GF insertion also destroyed membrane consistency.

Compared to the untreated GF composite, the morphologies of CEM composites with the pretreated GF present in Figure 10a,c,e,g show that the amounts of entwined matrices on GF’s surface increased. Figure 10b shows that the fiber surfaces became rough and had some a thin layer covered on fiber surface. Figure 10c,d show that the GFs wrapped by the polymer colloid layer completely and the polymer layer covered on GF became thicker than the GF in Figure 10b. This shows that much tighter connections were constructed due to 3-MPS modification. TES-treated GF were dispersed randomly under the thin layer of the polymer matrix (Figure 10e,f). The GFs were pulled out of the polymer matrix, and the surface cavities, which were created from the pulled-out glass fibers, were smooth. This indicates that the compatibility between the GF and matrix is weak, coinciding with the tensile strength and the DMA results. Figure 10g,h show that glass fibers were buried deeply under the membrane layer. There were some cross bridges on the interface between the individual fiber and matrix, indicating the strong connection between them.

The coupling agent acts as a bridge to connect the GF and copolymer matrix. When the composite material is subjected to stress, the stress can be transferred from the membrane matrix to GF and dispersed over it. The GF fillers can also inhibit crack expansion.

### 3.4. FT-IR Analysis

The FT-IR spectra of pure PE-g-PS copolymer and that with untreated GF and treated GF are shown in Figure 11. The right graph is the normalized FT-IR spectra of the left graph, showing the C-H stretching region between 1340 and 1420 cm^−1^. Compared to the composite with untreated GF, the new peaks appearing around 1060 cm^−1^ in 3, 4, 5 samples were due to asymmetric vibrations of (Si-O-Si), which were from the condensation between -OH in GF and the Si-OH groups in coupling agents. A new appearing peak at 1065 cm^−1^ in the spectrum of sample 6 was attributed to asymmetric vibrations of C-O-Si bonds, which originated from the condensation between -OH in GF and C-OH in PP-g-MA. Both emerging peaks were the evidence of a forming chemical bonding between GF and coupling agents. The peaks intensities around 3660 cm^−1^ in spectrum 3, 4, 5 and 6 amplified due to -Si-OH bonds from hydrolyzed 1,6 bis, 3-MPS, TES and PP-g-MA. In addition, the peaks around 2850–3000 cm^−1^ and 1400 cm^−1^ were attributed to C-H in methyl groups and C-H in aliphatic, respectively. The intermolecular hydrogen bonding among the GF, coupling agents and PE-g-PS matrix was characterized by these C-H stretching vibration. The C-H stretching vibration regions of 3,4,5,6 spectra presented on the right graph were shifted to the higher wavenumbers compared to the C-H spectra in pure PS-g-PE. This change in the C-H absorptions indicated the specific effect of hydrogen bonding between the matrix and coupling agents. Based on the above analysis, the possible reaction mechanism of GF, coupling agents and PE-g-PS copolymer is deduced as Scheme 2.

Since the mechanical properties are strongly related to the coupling agent’s concentration, the effects of concentration on formation of chemical bonding were investigated by the FT-IR spectrum. Figure 12 presents different samples’ FT-IR spectra with GF treated by increasing 3-MPS concentration. The characteristic peaks appearing at 1002 (Si-O-Si), 1130 (Si-O-Si) and 805 cm^−1^ (Si-O_3_CH_2_^−^) became sharpest when 3-MPS concentration reached 2%. Increasing 3-MPS concentration weakened corresponding peak intensities. The less sharp characteristic peaks indicate that lower amounts of Si-O-Si bonds generated between GF and coupling agents. The -C-H peaks attributed to methyl groups and aliphatic had the same trends as Si-O-Si peaks, indicating the more PE-g-PS could be attached on GF when GF was treated by 2% 3-MPS hydrolyzed solution. These results demonstrate that the 2% 3-MPS concentration has a superior modified effect on GF.

### 3.5. Effect of GF on Ion Exchange Capacity

The concern of adding fillers is that it may hinder ion transport and decrease ion exchange capacity. The result in Figure 13 shows that the CEM without GF had the largest IEC value, and reached 1.78 mmol g^−1^. The IEC of CEM composites with treated GF decreased slightly, 2–6% lower than CEMs without GF. The negative influence of adding GF on IEC value was in an acceptable range. However, the addition of untreated GF decreased the IEC value tremendously due to the vast emerging holes caused by GF insertion, which intercepted the ion transportation channels. The coupling agents could fill those vacancies, making the GF combine with the matrix tightly. The cations transported through aqueous membranes under the “diffusion mechanism”. Hydrated cation (H_3_O^+^) diffused through the aqueous medium in response to the electrochemical difference in this mechanism. The water connected cation (H^+^ (H_2_O)_x_) carried one or more water molecules transported through the membrane. Under this condition, the cations could bypass the obstacles and conduct through branch ionic groups of PE-g-PS membrane when encountering GF.

### 3.6. Effect of GF/CEMs Composite on Water Uptake and Swelling Rate

Adding GF into CEMs also affects water uptake and swelling rate due to the hydrophilic nature of GF. Figure 14 shows that the water uptake ability enhanced moderately after adding modified GF. The GF/CEM composite with 1,6 bis-treated GF had the highest water uptake. Accordingly, the swelling rates of composite (Figure 15.) with modified GF also increased due to a mass of water attracted by the GF. On the other hand, as fillers, the GF could hinder composite deformation to some extent. The deformation would occur only if the connection between GF and matrix is poor. For the blank sample, the swelling rate was relatively high (Figure 15) due to the absence of chemical bonding between the GF and composite matrix. Figure 14 and Figure 15 also indicate that the water uptake and swelling rate of all samples tended to be stable after immersing in 0.5 mol·L^−1^ NaCl for 2 h. Water uptake is a vital property since it positively correlates to IEC. However, excessive water uptake would result in a swollen membrane. Thus, the enhanced water uptakes of CEMs with 3-MPS, PP-g-MA and TES-treated GF lead to a higher IEC but increase the risk of unstable dimension.

## 4. Conclusions

This research provided information about interfacial adhesion and mechanical property improvement effects of four coupling agents on GF/CEM composite. The addition of modified GF enhanced the tensile strength and tensile modulus of GF/CEM composites but could not make composites suffer too much elongation. The interfacial adhesion between the matrix and GF, reflected by tan delta and the adhesive effect parameter, were improved with the addition of coupling agents due to the formations of Si-O-Si bonds between GF/coupling agent, and hydrogen bonding among the GF, coupling agents and matrix. The 3-MPS treated-GF showed satisfactory modification consequences, brought the most robust interface adhesion to the composite and led to the highest tensile strength. However, with increasing 3-MPS concentration to treat GF, the mechanical properties and interfacial adhesion worsened. The unmodified GF negatively influenced the composite’s mechanical properties due to the destruction of the membrane’s integrity and formation of micro-holes around the GF. The water uptake values increased after adding modified GF into CEMs. However, the IEC of GF/CEM composite decreased slightly. The swelling rate for the unmodified GF/CEM composite is relatively high due to the absence of chemical bonding between the GF and composite matrix.

This research provided a way of strengthening GF/CEM composite and pointed out which functional groups included in coupling agents could be useful to GF-reinforced composite. Future studies of GF-reinforced ion exchange membrane could use these conclusions as references. They might also consider other surface modifiers with similar characteristic structures or functional groups as effective coupling agents for GF-reinforced CEM composite.

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
