# Peer review of "Studies on Glass Fiber-Reinforced Poly(Ethylene-Grafted-Styrene)-Based Cation Exchange Membrane Composite"

_materials, 2020, doi:10.3390/ma13245597_

Round 1
Reviewer 1 Report
This review article (Manuscript number materials-1019422) presents information about the glass fiber reinforced poly(ethylene-grafted-styrene) based cation exchange membrane composite.
On the whole, the manuscript is not well-written and not logically arranged. The overall originality of the concept used here is medium-high. Therefore, I would recommend publication of this article in Materials on the condition a fundamental major revision of the manuscript will be carried out and the following points will be taken into consideration.
Detailed comments:
1. The abstract needs to be well written with future prospects of the work and describe in short the concept of preparation of glass fiber reinforced poly(ethylene-grafted-styrene) based cation exchange membrane (GF/CEM).
2. There is a lack of important references related to the e.g. glass fiber and poly(ethylene-grafted-styrene) materials. More detailed advantages of the present field must be mentioned in the Introduction. There should be a critical discussion of the state of knowledge of the field - what the previous studies establish definitively, what the tentative interpretations are, and what specific fundamental research questions still need to be studied.
3. The Introduction section appears to be a collection of data from research papers, however, the author's self-opinion is of importance while drafting a section of this type.
4. There is a lack of appropriate information about the copyright transfer agreement for each figure.
5. The conclusion reflects an overall summary of the field with further extension and includes future prospective - I would suggest clarifying this section.
6. Authors should work on editing text, spacing pause, Justify, fonts, italics, etc.
7. References should be adapted to the requirements of the journal.
8. The style and grammar leave much to be desired in many places, some parts of the text are difficult to understand.
After completing the above-mentioned corrections this work will be more readable. Therefore, it will be useful for the readers of the Materials.
Author Response
Thanks for your valuable report, please check the response file in the attachment.

Reviewer 2 Report
There are four coupling agents are used in this work named as [3-(Methacryloxy)propyl] trimethoxy silane (3-MPS), 1,6-bis (trimethoxysilyl) 13 hexane (1,6 bis), Poly(propylene-graft-maleic anhydride) (PP-g-MA) and Triethoxyvinylsilane (TES).
But PP-g-MA is a compatatbilizing agent instead of coupling agent.
There are only mention of 3-MPS mainly in the abstract and conclusion. It would be better is there are very short mention about other coupling agent as well.
Author Response

(The authors gave the same response as above.)

Round 2
Reviewer 1 Report
I would like to support this revised paper (Manuscript Number materials-1019422-peer-review-v2) for the publication in Materials.
All suggested changes were made (or discussed/clarified) by the authors.
The results are informative, and discussion is clear.
Summarizing, I think that this paper can be published as is.
This manuscript is a resubmission of an earlier submission. The following is a list of the peer review reports and author responses from that submission.
Round 1
Reviewer 1 Report
The article (Manuscript Number materials-748058) presents information about the improvement of interfacial adhesion between glass fiber (GF) and cation exchange membranes (CEM). In order to obtain desired properties, GF was modified by four kinds of coupling agents, which have effects on CEMs for enhancing tensile strength, tensile modulus, and interfacial adhesion. The additions of modified GF in CEM bring a positive influence on water uptake ability, but negative influences on ion exchange capacity.
Undoubtedly this research study has brought any interesting results. The authors did not present a specific theme in the article and the potential “real” application of such a procedure. The authors should be focused and make the target clear, the title and introduction need to be redone.
On the whole, the manuscript is not well-written, English could be improved in certain parts. The overall originality of the concept used here is not very high, therefore this method is not making the study useful for the respective field.
I do not recommend publication of this paper in Materials.
Reviewer 2 Report
The manuscript “The addition of coupling agents modified glass fiber into cation exchange membranes for enhancement of mechanical properties” is dedicated to an important problem of improving the thermal stability, mechanical strength and toughness of cation exchange membranes. The article considers the relevant properties of the membranes, such as tensile strength, tensile modulus, as well as ionic exchange capacity and water uptake of the reinforced membranes.
However, upon reading the manuscript, I have found multiple issues, and suggest that the authors address them:
1. The absence of the data for the membranes reinforced by glass fiber without modification, except for SEM image and the FT-IR spectrum, is striking. In my opinion, this is an important reference point for all of the studied samples. The authors should add the corresponding data or thoroughly explain why is it not available.
2. The data for 3-MPS suggests that the mechanical properties are sensitive to the concentration of the coupling agent. However, the concentrations of other coupling agents are not disclosed.
3. The authors speculate that the decrease of the mechanical strength of the membranes reinforced by fibers treated with 3-MPS is related to the saturation effect. To support the suggestion, the authors should provide the data not only for the saturation regime but also for lower concentrations of the coupling agent. The speculations on the mechanism behind the concentration-strength relationship should be corroborated by FT-IR analysis of the samples treated by different coupling agent concentrations.
4. Lines 230-232 contain the following passage: “It is speculated that 2% coupling agent is enough to hydrolyze Si-O of 20% GF addition. The supersaturated coupling agent would cover on modified glass fibers, decreasing the interfacial friction and adhesion between GF and membrane matrix.” At the same time, Section 2.2 states that the glass fibers were treated by the coupling agents before being mixed with the polymer, meaning that the results should not depend on the fraction of glass fibers in the composite and questioning the proposed explanation.
5. The authors should discuss and provide spectroscopic evidence for the possibility of hydrogen bonding between the glass fiber and the coupling agents.
6. Taking into account the heterogeneous composition of the samples, I would suggest that the authors provide the error estimates for their data or the values for the individual samples.
7. The lines 207-209 contain the following passage: “Compared with the blank sample, the crystallites transformation peaks of GF/CEM composites are shifting to a higher temperature range. It reveals the interfacial adhesion and compatibility between matrix and GF have been improved after being treated by coupling agents.”. First of all, it is unclear how such a comparison can be made given that the curve for “blank” sample in Figure 7 entirely lies below 100°C. In the second place, it should be clarified how the above-mentioned revelation can be made in the absence of data for the sample containing unmodified glass fiber, because at line 164 it is stated that the blank specimen is the membrane without the glass fiber.
8. The lines 212-213 contain the following statements: “The value of A is interface cohesiveness, which is an inverse proportion to the interfacial adhesion. The greater A, the better adhesion of the interface.” As can be seen from the preceding formula, and it’s derivation in the cited reference 19, the first statement is true and the second one is false, making the name “cohesiveness” obviously inappropriate for the above-mentioned factor A. Moreover, as one can see in reference 19, the derivation was performed under the assumption that the value of A is positive, so the values of A presented in Table 1 are nonsensical. At the same time, there’s no point in calculating the values, because calculating the volume fraction of the fibers from the data in Table 1 gives fiber content of 20% for all of the samples, so one could just compare the loss factors. It’s also worth noting that the volume fraction of the glass fibers is not explicitly presented anywhere in the article.
9. Lines 287-288 say: “The peak appearing at 1720 cm -1 in sample 5 responsible for C=O in COOH group. This is the evidence of condensation reaction between GF and PP-g-MA.” I think that clarification is needed how is it possible to distinguish between PP-g-MA grafted to the fiber and ungrafted PP-g-MA which also contains the COOH groups.
10. The introduction could benefit from citing some more articles on adhesion improvement between the glass fiber and a polymer membrane, such as Sanjay, M., & Yogesha, B. (2017); Křivčík, J., Neděla, D., & Válek, R. (2014); Somnuk, U., Yanumet, N., Ellis, J. W., Grady, B. P., & O’Rear, E. A. (2003); Li Huang, Hao Liu, Chengzhong Wang (2010).
11. There are multiple spelling and grammar issues in the manuscript. Mentioning all of the mistakes would result in a very long list, so here are only the examples from the Introduction: “and the later slightly decreased” (line 23, suggest “and the latter slightly decreased”), “there are renewed interests (line 28, suggest “there is renewed interest”), “have widely applications” (line 37, suggest “have wide applications”), “structure stability” (line 39, suggest “structural stability”), “high structure and chemical stability” (line 40, suggest “high structural and chemical stability”), “have attracted many attentions” (line 42, suggest “have attracted much attention”), “This hydroxyl groups weakens the adhesion” (line 48, suggest “These hydroxyl groups weaken the adhesion”), “was formed between the interface” (line 60, suggest “was formed at the interface”), “LUO Weica” (line 60, suggest “Luo Weica”), “to improve interfacial adhesions” (line 71, suggest “to improve the interfacial adhesion”), “was reported could effectively improve” (line 73, suggest “was reported to effectively improve”), “the TES did not treat on the GF directly” (line 74, suggest “GF was not directly treated by TES”), “the strengthen effect of their combination on GF/CEM composites” (line 81, suggest “the strengthening effect of their combination with GF/CEM composites”), “mechanical properties enhancement attributed to additive addition” (line 83, suggest “mechanical properties enhancement is attributed to the addition of the additives”).
It’s worth mentioning one more time that the list above is not complete. The other sections of the manuscript also contain lots of language mistakes, and the whole text must be proofread by someone with sufficient language knowledge.
12. There are also minor issues:
a) The curve designations in Figure 6 are confused.
b) The chemical formula of TES in Figure 1 contains a typo.
c) The abbreviations FE-SEM and FT-IR in the abstract are not explained.
d) In the list of sample characterization methods in Section 2.4 (lines 132-160) the names of the methods should be followed by full stops and not by commas.
e) At line 174 it is stated that “the CEM sample treated with 3-MPS has largest underneath area”, but the plot suggests that the area for the 3-MPS sample is lower than that for most of the other samples.
f) The order of reference citing is broken, with reference 20 cited before 18 and 19.
Summary:
Although the manuscript contains some original experimental results, it looks like important data is missing and the speculations about the mechanisms behind the discovered relationships are unfounded. This should be clarified by additional experimental data, which in my opinion could be obtained using the experimental techniques already presented in the manuscript.